# Mapping the evidence for patient and public involvement and engagement in statistical methodology research: A scoping review protocol

Naomi V. Bradbury[1,2], Julie Roberts[3], Selina Lock[4], Louise Haddon[1], Samina Begum[1], Beatriz Goulao[5,6], Nicola Mackintosh[1], Chris Newby[7], Michelle O'Reilly[8,9], Laura J. Gray[1,2,10,11] *

1 School of Medical Sciences, University of Leicester, Leicester, United Kingdom, 2 NIHR Leicester Biomedical Research Centre, University of Leicester, Leicester, United Kingdom, 3 SAPPHIRE Research Group, School of Medical Sciences, University of Leicester, United Kingdom, 4 Library and Learning Services, University of Leicester, Leicester, United Kingdom, 5 Aberdeen Centre for Evaluation, University of Aberdeen, Aberdeen, United Kingdom, 6 College of Medical, Veterinary and Life Sciences, University of Glasgow, Glasgow, United Kingdom, 7 Medical School, University of Nottingham, Nottingham, United Kingdom, 8 School of Criminology, Sociology and Social Policy, University of Leicester, Leicester, United Kingdom, 9 Leicestershire Partnership NHS Trust, Leicestershire, United Kingdom, 10 NIHR Applied Research Collaboration East Midlands, University of Leicester, Leicester, United Kingdom, 11 Leicester British Heart Foundation Centre of Research Excellence, University of Leicester, Leicester, United Kingdom

* lg48@le.ac.uk

## Abstract

### Objective

To identify and map the existing literature on the conduct of patient and public involvement and engagement (PPIE) for statistical methodology.

### Introduction

PPIE refers to the consideration of patient and public perspectives into research development, conduct and dissemination and, while commonly integrated in applied healthcare research, it currently remains underutilised in statistical methodology research. Many statistical methodologists lack confidence in conducting PPIE citing barriers such as insufficient training, unclear tasks, and concerns about impact.

### Inclusion criteria

This review will examine literature on PPIE in the context of statistical methodology aiming to inform the design or analysis of healthcare research studies and related methodological research domains such as trials methodology and health data science.

**Data availability statement:** No datasets were generated or analysed during the current study. All relevant data from this study will be made available upon study completion.

**Funding:** This study is funded by the Medical Research Council (grant number UKRI1488). This study is supported by the National Institute for Health and Care Research (NIHR) Applied Research Collaboration East Midlands (ARC EM) and Leicester NIHR Biomedical Research Centre (BRC). LG is an NIHR Senior Investigator.

**Competing interests:** The authors have declared that no competing interests exist.

## Methods

A three-step search strategy, developed with an information specialist and based on JBI guidelines, will identify published and unpublished PPIE literature. This will include database searches, hand-searching key journals, and grey literature searches via search engines. The authors will contact their professional networks to identify additional material. The literature will be screened, selected and extracted by two independent researchers. Results will be presented in an evidence map and using a qualitative, thematic analysis.

## Introduction

Patient and public involvement (PPI) can be defined as 'research that is developed with or by the public' and is an integral part of healthcare research [1]. Individuals with lived experience of health conditions are often best placed to inform researchers about the needs of patients and, therefore, the involvement of patients in all aspects of the research process for health studies can improve the quality and relevance of the results [2]. Public engagement is different to public involvement and is the process of disseminating research findings in a way that is comprehensible to the lay person. Therefore, we will refer to patient and public involvement and engagement as PPIE throughout this protocol to encompass both aspects of integrating the public into the research process.

Statistical methodology research involves the development, evaluation or comparison of statistical methods for the design or analysis of research studies. For this review, we are interested in the statistical methodology research which intends to improve the design and/or analysis of applied healthcare research.

There is extensive literature on best practice for PPIE in applied health research [3–5]. In contrast, there is no guidance on how to conduct meaningful PPIE for statistical methodology research. The technical nature of this research and the often-confusing terminology used, can make PPIE challenging. This can lead to tokenistic input. Tokenism occurs when the skills and experiences of public contributors are underestimated, or when they are not engaged sufficiently to allow meaningful input to the research [6]. While tokenism can occur in all areas of health research, the risks in statistical methodology research are greater due to the effort and creativity needed to make statistical methodological concepts comprehensible to public contributors.

In 2023, a survey of statistical methodologists was conducted to establish current PPIE practices and attitudes [7]. Of 119 respondents, over 80% did not feel confident in carrying out PPIE for statistical methodology research. Perceived barriers included: the technical complexity of the research; choice of suitable tasks for meaningful contribution; perceptions that PPIE activities could not realistically have an impact; and lack of training. When asked what would make them feel more confident about undertaking PPIE; training, guidance, and case studies were the top responses.

In developing an understanding of PPIE in statistical methodology research, a preliminary search of JBI Evidence Synthesis, the Cochrane Database of Systematic

Reviews, Epistemonikos, and PubMed was conducted for existing reviews on the topic. There have been recent reviews of PPIE in areas of non-applied healthcare research applications, for example, big data research [8], medical education [9], preclinical laboratory research [10] and personalised prevention using genomic data [11]. There are, however, no current reviews focusing on PPIE for statistical methodology research.

To address this, the objective of this scoping review is to identify and map the existing literature on the conduct of PPIE for statistical methodology research. We will include studies that have undertaken empirical research on how to conduct PPIE for statistical methodology research and examples or outputs from PPIE activities. From our preliminary scoping, it is apparent that relevant literature is not always described as statistical methodology by the authors. Therefore, we have used our knowledge of the field to incorporate the areas of statistical methodology that develop and evaluate statistical methods for healthcare (e.g., clinical trials methodology [12–15] and data science [16]).

This will allow us to learn how others have approached PPIE for studies of a methodological nature, what they have found to be the benefits and challenges, the impact of PPIE and any recommendations they have made. These findings will then be used to inform future work on the development of PPIE guidance for researchers and public contributors conducting statistical methodology research.

### Review question

What is the extent of the existing literature on the conduct of PPIE for statistical methodology and other closely related areas of methodological research?

- What have been the approaches and methods in existing examples of PPIE in statistical methodology research?

- What resources exist to aid the conduct of PPIE for statistical methodology research?

- What are the benefits and challenges of conducting PPIE for statistical methodology research?

- What is the impact of integrating PPIE into statistical methodology research?

- How has PPIE been evaluated in statistical methodology research?

- What recommendations have been made for future PPIE to inform statistical methodology research?

## Materials and methods

This protocol has been developed following the JBI Manual for Evidence Synthesis guidance for scoping reviews and using their Population, Concept, Context framework for inclusion criteria [17] and the scoping review will be reported using the PRISMA Extension for Scoping Reviews (PRISMA-ScR) [18]. This protocol has been written according to the PRISMA-P guidelines and a completed checklist is given in S1 Appendix. The study commenced in August 2025, with searching, selection and data extraction expected to be completed by early 2026. It is expected that the results from the review will be available by mid 2026.

### Inclusion criteria

**Population.** There are no restrictions on the population.

**Concept.** The review will consider literature on the conduct of PPIE. We will use the NIHR description of public involvement in research as "research being carried out 'with' or 'by' members of the public rather than 'to', 'about' or 'for' them" [19]. We will also include studies which consider engagement in the context of PPI, which the NIHR define as 'activity where information and knowledge about research is provided and disseminated' [20]. However, terminology around PPIE is still evolving and is not internationally standardised so other phrases that encompass public involvement such as "co-production", "community participation" or "patient engagement" will also be eligible for inclusion [21].

**Context.** We will define statistical methodology research as the development, evaluation or comparison of methods for designing studies, analysing data, and presenting results. Here we are specifically interested in statistical methodology research which aims to improve the conduct of applied health and care research studies.

We will be broad and inclusive in our definition of statistical methodology research for applied healthcare and consider the following (non-definitive) list of areas of quantitative healthcare methodological research to be part of that definition: observational studies, clinical trials, health technology assessment, health data science, mathematical modelling and simulation studies.

**Types of sources.** We will include both published peer reviewed articles and grey literature. We anticipate, from our expertise in the discipline, that there will be relevant sources of information which are not published in peer reviewed journals (e.g., websites, blogs or webinars).

We will restrict articles to those published from 1996 onwards as this was when the Department of Health in England established a Standing Advisory Group on 'Consumer Involvement in the NHS Research and Development Programme', which represents the beginning of what we now call PPIE [22].

## Exclusion criteria

Any literature on applied patient research (e.g., clinical trials or observational studies) where the PPIE is informing the methods to be used in such studies rather than being of a methodological nature will be excluded as will any non-human studies (e.g., preclinical or animal studies). Methodological studies pertaining to the development, evaluation or comparison of qualitative methods will be excluded. Any literature where we are unable to access the full text (i.e., conference abstracts) will be excluded as they will not contain enough information for data extraction.

We will exclude studies where the PPIE is informing a study that does not come under our definition of statistical methodology although this distinction can be challenging to describe. We would not, for example, include a study that reports on the PPIE approach to choosing a randomised controlled trial design to be used to answer a particular applied research question, but we would include a study of PPIE to inform the development of a new type of randomised controlled trial design.

For example, Cro *et al*, a study discovered during our initial scoping of the literature that met our inclusion criteria, conducted some PPIE work alongside methodological research on the use of estimands in clinical trials. Here, public partners co-developed a tool to explain what estimands are and explored how estimands can be discussed during trial design [15].

**Search strategy.** A three-step search strategy will be conducted following the JBI recommendations with the aim to locate both published and unpublished literature. The search strategy for the first and second searches were developed in collaboration with an information specialist.

In the development of this protocol, the first step of the search strategy involved an initial search of MEDLINE and Scopus to identify articles on the topic. A validated filter was used to identify PPIE articles [23]. We made some adaptations to the PPIE search filter to include the following additional relevant terms: carer, co-production, co-development and co-creation. Keywords in the titles and abstracts and the MeSH terms of the relevant articles identified in this first step were used to adapt and refine the search for the second step (S2 Appendix).

The finalised search filter will be used in the second step of the search strategy, with the search filter adapted for selected databases judged to be most relevant to the topic (MEDLINE (Ovid), CINAHL (EBSCOHost) and APA PsycInfo (Ovid)). Given the lack of consistent terminology for both PPIE and statistical methodology for database searching, we anticipate additional search methods will be important to find relevant literature.

We will hand search the following journals and websites that we have identified as being highly relevant to this review:

• BMC Research Involvement and Engagement

• Journal of Participatory Medicine

- Health Expectations

- Research for All

- https://www.learningforinvolvement.org.uk/

To identify unpublished or other hard-to-find literature, Google, DuckDuckGo and Mednar will be searched in incognito mode to remove the influence of past search histories and, in the case of DuckDuckGo, with location set to 'all regions'. We will conduct our searches using combinations of terms for both PPIE and statistical methodology, e.g., "Co-production" AND "statistical methodology", "PPIE" AND "Data Science" (S2 Appendix). This will result in many searches, therefore, we will restrict our findings to the first 20 results to maximise search efficiency. We will contact our network of statisticians involved in PPIE (PPIE for Statisticians Community of Practice) to identify additional relevant literature or organisational websites. All types of grey literature (including non-textual resources such as videos) will be considered as long as there is sufficient detail about the PPIE to allow for meaningful data extraction.

The third step of the search will be to screen the reference lists of all studies selected for data extraction to ensure no relevant studies have been missed.

**Evidence selection.** Following the search, all identified sources will be imported into Covidence [24] and duplicates removed. Pilot testing of both title and abstract screening and full text screening will take place using 25 randomly selected articles. The team will screen these using the eligibility criteria and then meet to discuss discrepancies and make any modifications as required.

Following pilot testing, title and abstract screening and full-text screening will both be performed by two reviewers independently with a third reviewer deciding when there are any disagreements. A PRISMA flow diagram of the process will be produced outlining the number of articles at each stage as well as the reasons for exclusions [25].

**Data extraction.** Data extraction will be conducted using both Covidence and NVivo [26]. In Covidence, structured data will be extracted by two independent reviewers with a third reviewer resolving any disagreements. During the development of the data extraction form it will be pilot tested on three articles and modifications will be made on an iterative basis as needed as the data extraction proceeds. The final data extraction form will be reported in the scoping review.

The data extraction form will capture key information including: year of publication, country of origin, study funding, category of medical statistics methodology research, PPIE methodology, number of PPIE participants, participant remuneration, participant acknowledgement and diversity of PPIE members. Study-level risk of bias for published papers will be assessed using the CASP Qualitative Studies Checklist [27].

Using NVivo, we will extract narrative data on the objectives, approaches, benefits, challenges, evaluation, impacts, resources, key findings, recommendations and evidence gaps for PPIE for statistical methodology research.

## Data analysis and presentation

Data will be charted to provide a narrative overview of the characteristics of included studies, focusing on the key themes related to the research questions. Across statistical methodology research and other related areas, the synthesis will:

- Discover the methods and approaches used for PPIE

- Summarise the benefits to health research of conducting PPIE

- Outline any challenges of conducting PPIE for methodological research

- Detect how PPIE has been evaluated in methodological research

- Identify the impacts of integrating PPIE into methodological research

- Identify available resources for supporting PPIE

- Provide recommendations for future research and practice

- Identify research gaps

Due to the nature of the research question, meta-analysis will not be conducted but we will reflect upon possible sources of meta-bias and limitations in the discussion section of the resulting article. Upon completion, our team and PPIE group will convene to review the findings. During an externally facilitated meeting, we will produce an evidence map [28]. This visual map will outline the available evidence, areas for future research, and feed into the development of guidance for conducting PPIE in this area.

**Patient and public involvement.** This review is the first part of a larger programme of work which aims to develop guidance for conducting PPIE to inform statistical methodology research. Our PPIE co-applicant, who is part of the research team and has reviewed this manuscript, will chair a PPIE advisory group. This advisory group will meet three times a year (regular monthly updates in the form of 'PPIE postcards' will be sent in between meetings) and will advise on all aspects of the project, including this review. Our PPIE input has informed this scoping review, suggesting that both the positive and negative experiences of PPIE in this area should be collected and the importance of highlighting how the research, either directly or indirectly, will ultimately make a difference to people. Our PPI co-applicant and Patient Advisory Group will be involved with the co-development of a visual evidence map, attending the externally facilitated meeting.

**Ethics and dissemination.** Ethical approval is not required for this review as it will use existing published aggregate data only.

The results of the review will be published in a peer reviewed journal and, in addition, will be shared through our social media accounts on LinkedIn and Bluesky.

## Discussion and conclusions

While PPIE is well-established in applied health research, its integration into statistical methodology remains limited and underexplored. This scoping review aims to synthesise the evidence on how PPIE has been conducted for statistical methodology research, the challenges and benefits encountered, and the resources and recommendations available to support meaningful involvement. By mapping the current landscape, this review will provide a clearer understanding of how PPIE can be effectively integrated into statistical methodology research.

This protocol describes a comprehensive search strategy that includes both peer-reviewed and grey literature. One limitation is that the evolving and inconsistent terminology surrounding PPIE and statistical methodology may pose challenges in identifying all relevant literature. We have chosen to take a "broad and shallow" approach to our grey literature search to incorporate many common terms for both "statistical methodology" and "PPIE" into the search. We acknowledge that this means we may not find some literature that is not highly ranked by search engines. We will aim to maximise the identification of relevant literature by citation searching included studies and contacting our professional PPIE network.

The findings of this review will inform the development of practical guidance for researchers and public contributors seeking to conduct PPIE within statistical methodology research and contribute to a more nuanced understanding of how to promote meaningful and impactful PPIE in this field.

## Supporting information

**S1 Appendix. Completed PRISMA-P checklist.**
(DOCX)

**S2 Appendix. Search strategy.**
(DOCX)

## Author contributions

**Conceptualization:** Laura Gray.

**Funding acquisition:** Laura Gray.

**Methodology:** Naomi V Bradbury, Laura Gray, Selina Lock.

**Project administration:** Laura Gray.

**Supervision:** Laura Gray.

**Writing – original draft:** Naomi V Bradbury, Laura Gray.

**Writing – review & editing:** Julie Roberts, Selina Lock, Louise Haddon, Beatriz Goulao, Nicola Mackintosh, Chris Newby, Michelle O'Reilly, Samina Begum.

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
