## [Decision Letter · Decision Letter 0]

16 Sep 2025

Dear Dr. Gray,

We look forward to receiving your revised manuscript.

Kind regards,

Rebecca F. Baggaley

Academic Editor

PLOS ONE

Journal Requirements:

“This study is funded by the Medical Research Council (grant number UKRI1488). This study is supported by the National Institute for Health and Care Research (NIHR) Applied Research Collaboration East Midlands (ARC EM) and Leicester NIHR Biomedical Research Centre (BRC). LG is an NIHR Senior Investigator.”

Reviewers' comments:

Reviewer's Responses to Questions

**Comments to the Author**

1. Does the manuscript provide a valid rationale for the proposed study, with clearly identified and justified research questions?

Reviewer #1: Yes

2. Is the protocol technically sound and planned in a manner that will lead to a meaningful outcome and allow testing the stated hypotheses?

Reviewer #1: Partly

3. Is the methodology feasible and described in sufficient detail to allow the work to be replicable?

Reviewer #1: No

4. Have the authors described where all data underlying the findings will be made available when the study is complete?

Reviewer #1: Yes

5. Is the manuscript presented in an intelligible fashion and written in standard English?

Reviewer #1: Yes

You may also provide optional suggestions and comments to authors that they might find helpful in planning their study.

Reviewer #1: I thank the editors for inviting me to review this protocol. The authors address an important topic and clearly explain how this review aims to fill a gap in the existing literature. I look forward to reading the results once available. I do, however, have several notes regarding the protocol.

1. It is unfortunate that this protocol is being published after the study has already commenced in July and is expected to be completed by September (this month). This timing reduces the function of the paper as a tool for transparency and pre-study registration, and also limits the opportunity for review comments to help strengthen the work. I did not find information on whether the protocol was published elsewhere before the start of the study; This should be clarified. The timing of the protocol submission should also be made clear in the manuscript. Nevertheless, I provide my comments below.

2. Regarding inclusion: Is patient and public involvement in the design of applied studies considered eligible? For instance, if PPI is integrated at all stages of an empirical health related study, including design, and the experiences are documented, would such a paper qualify for inclusion, even if it is not a methodological paper on its own? If a paper includes a patient or lay person as co-author, but does not explicitly describe their involvement, will it still be considered for inclusion?

3. When discussing quantitative healthcare methodological research, the authors mention clinical trials, health data science, mathematical modeling, and simulation studies. Does this also include the design of observational cohort studies, case studies, or the statistical design of analyses for qualitative studies?

4. The authors intend to include grey literature, but the protocol lacks detail on the standards, limitations, or formats applied to this category. For example, would a folder, a video recording, or an opinion blog be eligible? What would not be?

5. Based on these points, I feel the methodology is not yet explicit enough to guarantee transparency or support replication. Greater detail should be provided on study inclusion and exclusion criteria and the reasons for these decisions. A PICOS framework could help structure this, as could examples of papers that would qualify for inclusion.

6. The authors restrict the search to papers published after 1996, when the advisory group on consumer involvement was established, describing this as a milestone in modern PPIE. However, they do not explain why earlier studies are considered irrelevant. Why does this milestone justify excluding earlier literature?

7. The authors have limited grey literature sources to the first 20 results. What is the rationale for this cutoff? Do they have supporting evidence? For comparison, Wichor Bramer et al. examined the yield of additional references within the first 100–200 results of Google Scholar searches, and their findings could serve as guidance for setting such a limit.

8. On page 9, line 234, the authors state they “anticipate” that the data extraction form will capture certain information. In the context of a review protocol, however, data items to be extracted should be prespecified (as required in registries such as PROSPERO). Any additional extraction should later be reported as deviations from the original protocol. Simply “anticipating” extraction items risks later omission of items that yield unfavorable results, which could introduce bias or raise concerns of cherry-picking. Prespecification is therefore an important transparency safeguard.

9. On page 10, under patient and public involvement: the PPIE group is described as advising the research group. Given the nature of this study, it would be helpful to clarify the extent of their role. Are they solely advisory (with recommendations that may be ignored), or are they members of the research team? Do they contribute to manuscript drafting and revision, or do they remain external advisors?

10. I disagree with the statement under PRISMA-P that risk of bias is “not applicable.” Bias can arise in all forms of research, whether qualitative or quantitative, systematic or scoping. The authors should reflect on the possible sources of bias and limitations in this review, including when narrative synthesis is applied. For example, are included studies written by groups with a vested interest?

11. In the data collection framework, I would encourage the inclusion of two topics frequently discussed in patient communities: (1) remuneration and acknowledgment of patient contributions, and (2) diversity of representation within PPI. Capturing such information would add value to the scientific community.

**Do you want your identity to be public for this peer review?** For information about this choice, including consent withdrawal, please see our Privacy Policy

Reviewer #1: **Yes: ** Stijntje Dijk

---

## [Author Response · Author response to Decision Letter 1]

13 Oct 2025

We have uploaded a response to the reviewer's comments file containing a detailed response to each of the reviewers points with references to where these changes have been made in the manuscript, alongside updated manuscript and supplementary information files.

---

## [Editor Report · Decision Letter 1]

15 Oct 2025

Mapping the evidence for patient and public involvement and engagement in statistical methodology research: a scoping review protocol

PONE-D-25-34387R1

Dear Dr. Gray,

We’re pleased to inform you that your manuscript has been judged scientifically suitable for publication and will be formally accepted for publication once it meets all outstanding technical requirements.

Kind regards,

Rebecca F. Baggaley

Academic Editor

PLOS ONE
---

## [Editor Report · Acceptance letter]

PONE-D-25-34387R1

PLOS ONE

Dear Dr. Gray,

I'm pleased to inform you that your manuscript has been deemed suitable for publication in PLOS ONE. Congratulations! Your manuscript is now being handed over to our production team.

Kind regards,

on behalf of

Dr. Rebecca F. Baggaley

Academic Editor

PLOS ONE